# Study of Mineral Composition and Quality of Fruit Using Vascular Restrictions in Branches of Sweet Cherry

**DOI:** 10.3390/plants12101922

**Published:** 2023-05-09

**Authors:** María Paz Quiroz, Víctor Blanco, Juan Pablo Zoffoli, Marlene Ayala

**Affiliations:** 1Departmatento de Fruticultura y Enología, Facultad de Agronomía e Ingeniería Forestal, Pontificia Universidad Católica de Chile, Santiago P.O. Box 78204360, Chile; 2Department of Horticulture, Tree Fruit Research and Extension Center, Washington State University, Wenatchee, WA 98801, USA

**Keywords:** xylem, phloem, calcium, source–sink, CHO, girdling, fruit firmness

## Abstract

Calcium (Ca) and carbohydrate (CHO) supply in sweet cherry have been associated with fruit quality at harvest and during storage. There is little published information integrating CHO and Ca availability and distribution in sweet cherry and their effects on fruit quality. Accordingly, in the 2019–20 season, vascular restrictions were imposed on the phloem (girdling, G, stopping phloem flow) and xylem (transverse incision, S, cutting 50% of xylem cross–section area) of individual vertical branches of the sweet cherry combination ‘Lapins’/Colt trained as Kym Green Bush system to modify mineral and CHO composition in fruit and associate such changes with quality at harvest and storage. The girdling to the phloem was used to induce changes in CHO distribution. The transverse incision to the xylem was a tool to modify Ca distribution. Five treatments (TR) were implemented: TR1–CTL = Control (without vascular restriction), TR2–G, at its base, TR3–G + G: at its base, and G further up at the change of year between the second and the third years of growth TR4––S and TR5–S + G. The vegetative (i.e., shoot and leaf growth), reproductive (i.e., fruit set and yield) development and stomatal conductance were monitored. Each branch was divided into the upper (1–and 2–year–old wood) and the lower (3–and 4–year–old wood) segments of the restriction applied. The quality and mineral composition (Ca, Mg, K, and N) of fruit borne on each segment were measured at harvest. The upper segment of TR3–G + G branches were harvested 10 d before the lower segment. The fruit from the upper segment of TR3–G + G was the largest, the sweetest, and had the higher titratable acidity concentration. However, fruits of this segment were the softest, had the lowest Ca concentrations, and had the highest ratios of N:Ca and K:Ca, compared with the other TRs. TR3–G + G branches developed the highest number of lateral current season shoots including shoots below the second girdling in the lower segment of the branch. This vegetative flow of growth would explain the mineral unbalance produced in the fruit from the upper segment of the branch. TR2–G did not register changes in fruit quality and mineral concentration compared with TR1–CTL. Surprisingly, the fruit from the branches with xylem restriction did not show changes in Ca concentration, suggesting that the xylem stream was enough to supply the fruit in branches without lateral shoot development. Fruit firmness was positively related to fruit Ca concentration and negatively related to the ratios of K:Ca and N:Ca.

## 1. Introduction

Calcium (Ca) concentrations in fruit have been associated with high–quality fruit in sweet cherry [1]. Calcium plays several important roles as a component of the cell wall, membrane stabilizer, and a secondary messenger [2,3,4,5]. Between 65 and 70% of the Ca of fruit is in the cell wall [5]. Calcium is involved in the stabilization and control of membrane permeability and, signaling to activate several cellular processes [6,7]. 

Calcium is a divalent cation that is immobile in the phloem [8] and so differs from other macronutrients such as nitrogen (N), phosphorus (P), potassium (K), and magnesium (Mg), which all have high phloem mobility [9,10]. Long–distance Ca transport in plants occurs exclusively in the xylem, where sap flow is driven among the various organs of a plant, along a gradient of hydrostatic pressure generated by a difference in water potential at the two ends [8,11]. Most of the Ca requirements of the various plant organs are served by the mass flow of free Ca ^+ 2^ in the xylem. The higher the transpiration rate, the more water is carried in the xylem, and more Ca will be transported to fruit [12,13]. In apple, the transpiration of leaves plays promote Ca inflow to the fruit [14] and xylem restrictions might negatively affect Ca flow to actively growing sinks, such as young fruits [1,15]. Calcium deficiencies induce physiological disorders in fruit [16] such as bitter pit in apple (*Malus domestica*) [17], black end in pear (*Pyrus pyrifolia*) [18], and blossom end rot in tomato (*Solanum lycopersicum*) [19].

Calcium uptake from the soil solution is promoted by the higher transpiration rates of leaves, compared to fruits [1,7,20,21], which results in much higher Ca contents in the leaves than in the fruit [19]. During the first stages of fruit development in apple [22], grape (*Vitis vinifera*) [23], kiwifruit (*Actinidia deliciosa*) [24], and sweet cherry [25,26], Ca is transported to the fruit through the fully active xylem vessels. However, as fruit enters the ripening stage, the fruit xylem loses its functionality, which reduces and eventually stops Ca uptake by the fruit. Meanwhile, the still—active phloem tissues are mainly responsible for the supply of water to the fruit, along with sugars and other nutrients [21,25,27,28]. In sweet cherry, xylem inflow decreases during maturation, with dysfunction progressing from the bundles at the distal (stylar) end to those at the proximal (stem) end [29]. A continuous decrease of Ca concentration in sweet cherry occurs from 14 days after full bloom (DAFB), being the result of a steady increase in fruit volume with little or no import of new Ca—a simple dilution effect by growth expansion [30]. On the other hand, vigorous shoot growth and high leaf area to fruit (LA:F) ratio also influence the final Ca concentration in apple fruit [31], spur leaf area affects Ca uptake by apple fruit, since the removal of 22% of primary leaf area at bloom reduces Ca content in fruit by 9% [14]. In kiwifruit, low fruit transpiration is correlated with reduced Ca concentrations at harvest [24].

Calcium applications during fruit development have positive effects on fruit firmness [26,32,33,34] and reduction of fruit cracking in sweet cherry fruit [35]. In addition, CHO availability during sweet cherry fruit development is required to achieve high–quality fruit; source limitations during fruit development reduce fruit quality in sweet cherry.

The main carbohydrate (CHO) source for sweet cherry fruit development are the photoassimilates synthesized by its three types of leaf populations: fruiting spur, non–fruiting spur, and current–season shoot (CSS) leaves [36,37]. The contribution of each leaf population as a source of photoassimilates varies during fruit growth [38], generating competition between actively growing aerial sinks (i.e., fruit and CSS) [37,38]. The phloem is the pathway for CHO transport and distribution from leaves and storage reserves to developing aerial sinks (i.e., flowers, fruits, and CSS) in sweet cherry [39]. When sources of photoassimilate (i.e., leaves and storage reserves) are insufficient to supply sink demand of fruit, buds, spur, CSS, wood, and roots, negative effects are observed in fruit quality, particularly size, TSS, firmness, and postharvest life [37,39]. An optimal LA:F ratio is a key factor to ensure an adequate balance between fruit quality and vegetative growth in sweet cherry [39,40]. Girdling (i.e., interruption of phloem translocation) has been reported as a horticultural practice that advances maturity without reducing fruit quality in peaches (*Prunus persica*), plums (*Prunus cerasifera*) and sweet cherries [41,42,43,44]. Some studies using girdling in sweet cherries have elucidated the fate of current photosynthates produced by different leaf populations [36,44]. On the other hand, xylem restrictions [15] might reduce Ca concentrations in fruit due to changes in transpiration rates, however, not enough information on sweet cherry is available.

No information about the effect of girdling as a research tool to elucidate the impact of Ca and CHO distribution on fruit quality and storage in the Kym Green Bush training system has been published. For this reason, this study aimed at integrating Ca and CHO availability and their effects on fruit quality and the postharvest life of sweet cherry in KGB. Accordingly, we hypothesized that phloem and xylem restrictions on individual vertical branches would modify Ca concentrations and photoassimilate content in fruit, which in turn, would affect sweet cherry quality. To test this, girdling and xylem cuts were implemented to induce changes in Ca and CHO distribution among growing sinks, particularly towards fruit.

## 2. Results

### 2.1. Harvest Date and Yield

The vascular restrictions imposed on ‘Lapins’/Colt sweet cherry vertical branches at eight DAFB had significant effects on the harvest date and fruit yield (Figure 1). Fruit from branches without vascular restrictions (TR1–CTL) were harvested at 88 DAFB as well as fruits from branches with phloem girdling at their base (TR2–G). In contrast, fruits from the branches with restriction in the xylem flow (TR4–S) were harvested two days later (90 DAFB) than fruit from the control (TR1–CTL). Fruits from branches with restriction in the xylem flow and phloem girdling at their base (TR5–S + G) had an even longer fruit development period, being harvested five days later (93 DAFB) than fruit from branches without vascular restriction (TR1–CTL) and three days later than branches with cutting 50% of xylem cross–section area (TR4–S).

The branches with two girdling cuts (TR3–G + G,) were the only ones with significant differences in fruit phenology and harvest date as the fruiting segments (i.e., upper or lower half) within the same branch were compared. Fruits from the lower segment (0 to ~1.2 m) of TR3–G + G had a longer fruit development period being harvested 10 days later (93 DAFB) than the upper part of the branch (~1.2 to 2.4 m) and the latter five days earlier (83 DAFB) than those from the control branches (TR1–CTL).

The control branches without vascular restriction (TR1–CTL; i.e., upper segment + lower segment = 749 + 826 g branch^−1^) and those with a sectional cut of the xylem (TR4–S; 724 + 555 g branch^−1^) had significantly higher total fruit yields per branch than the TR3–G + G (749 + 256 g branch^−1^) and TR5–S + G (604 + 409 g branch^−1^) branches, which had the lowest yields per branch (Figure 1). 

The fruit yield from the lower segment of control branches (TR1–CTL) was significantly higher (*p* = 0.001) than the fruit yields observed for the same segment in TR3–G + G and TR5–S + G branches (Figure 1). The fruit yield from the upper segment did not have significant differences among treatments. On the other hand, only the branches with two girdling cuts (TR3–G + G) showed significantly (*p* = 0.003) higher yields on the upper segment than the lower one, with twice as many fruits on the upper section of the branch (Figure 1). 

### 2.2. Leaf Area and LA to F Ratios

The vegetative growth of branches was influenced by the vascular restrictions imposed in early spring (8 DAFB) (Table 1). The branches with double girdling (TR3–G + G), without vascular restrictions (TR1–CTL), and with girdling and xylem restriction at the base (TR5–S + G) registered the highest total LA at harvest. The number of spurs per branch among treatments was similar, but spurs from TR1–CTL and TR3–G + G branches had significantly higher LA (230 to 275 cm^2^/spur). Besides, branches with double girdling (TR3–G + G) showed a higher number of CSS (5 CSS/branch) compared to the other treatments (Table 1).

Differences in the LA and number of fruits per branch led to significant differences in the LA:F ratio among treatments. (Table 1). The lowest number of fruits per branch was observed in branches with two girdling cuts (TR3–G + G), while the branches without vascular restriction (TR1–CTL) obtained the highest number of fruits (Table 1). The branches with two girdling cuts (TR3–G + G) exhibited the highest LA:F ratios, followed by TR5–S + G (Table 1). In these branches, the higher LA per branch was caused by larger spur leaves and higher CSS number below the girdling between the 2–and 3–year–old wood.

### 2.3. Fruit Quality at Harvest

The fruit fresh unitary weight was positively influenced in the branches with two girdling cuts (TR3–G + G). In these branches, the fruit from the upper segment had the significantly highest (*p* = 0.0001) average fruit fresh weight (FW, 11.7 g fruit^−1^; Table 2) compared to the other treatments (from 8.1 to 9.8 g fruit^−1^; Table 2). On the other hand, the fruit FW from the lower segments showed no significant differences among treatments (overall average of 10.3 g fruit ^−1^).

Regarding the total fruit dry weight (DW), the branches with a cross–sectional cut to the xylem and girdling at their base (TR5–S + G), had lower fruit DW values (*p* < 0.0001), in both upper and lower segments, compared to the other treatments, which had similar values (Table 2). The branches from TR–5–S + G had a lower total fruit DW compared to the CTL branches (TR1–CTL). The lower segment of the branches with two girdling cuts (TR3–G + G) showed significantly lower total fruit DW than the other treatments and was significantly lower than the total fruit DW of the upper segment (Figure 2).

The fruit diameter was significantly (*p* < 0.0001) affected by the vascular restrictions imposed (Table 2). The fruits from the upper segment of the TR3–G + G and branches without vascular restrictions (TR1–CTL) were significantly (*p* < 0.0001) larger (i.e., 28.9 to 29.5 mm) than the fruit from the treatments with xylem restrictions TR4–S (i.e., 27.0 mm) and TR5–S + G (i.e., 28.0 mm). Significant differences in the lower segment of branches were observed, but without a clear trend among treatments. However, the branches with a cross–sectional cut and girdling at their base (TR5–S + G) had smaller fruit sizes than branches without vascular intervention in upper and lower segments (Table 2).

The firmness of fruits from the upper segment of branches was significantly different (*p* = 0.04) among treatments. Fruits from branches with double girdling (TR3–G + G) showed a significantly lower firmness (67.5 durofel unit) than fruits (73.2 durofel unit) from branches with only girdling at the base (TR2–G). However, no significant differences (*p* = 0.06) in fruit’s firmness (71.5 durofel unit) were detected in the lower segment of branches, probably associated with the high variability among treatments (Table 2).

The TSS and TA of fruits on the upper segment of branches were significantly (*p* < 0.05) influenced by vascular restrictions. The branches with two girdling cuts (TR3–G + G) had the highest TSS (18.6%) and TA (1.02%) values, followed by TR1–CTL. The fruit from the lower segment did not present significant differences for TSS and TA, among treatments (Table 2). Regardless of the treatment, fruits from the upper segment had significantly higher TSS values than fruits from the lower segment (Table 2).

### 2.4. Water Relations

During Stage III of fruit development, stomatal conductance (gs) at noon ranged between 185 and 210 mmol m^−2^ s^−1^ and no significant differences (*p* > 0.05) were observed among treatments. Moreover, gs was highly and negatively correlated with VPD (Figure 3, R^2^ = 0.6, *p* < 0.05). In general, for all treatments, stomatal closure increased (i.e., gs values decreased) when VPD reached values above 2 kPa.

In the branches with 50% restriction in the xylem (TR4–S), gs decreased to 120 mmol m^−2^ s^−1^ when VPD values were higher than 1.1 kPa; while for the same environmental conditions, the gs values in TR1–CTL were 62% higher (190 mmol m^−2^ s^−1^). These results indicate that under low atmospheric water demand, the restrictions in the xylem did not affect gas exchange; however, under medium atmospheric water demand conditions the branches with 50% restriction in the xylem (TR4–S) suffered stomata closure, which might affect net photosynthesis.

### 2.5. Fruit Mineral Composition and Ca Relationships

The vascular restrictions did not significantly affect the mineral content of fruits–Ca, N, K, and Mg, expressed as mg per fruit (Table 3), in the upper and lower segments of vertical branches. However, fruit from the upper segment of the branches with double girdling (TR3–G + G) did have a significantly lower concentration of Mg (expressed as mg per 100 g^−1^ fruit). In addition, there were significant differences in the N:Ca (*p* = 0.040) and K:Ca (*p* = 0.027) stoichiometric ratios of fruits from the upper segment of branches among treatments. The fruit from TR3–G + G registered a trend to higher N:Ca and K:Ca values compared to the other treatments (Table 3).

In general, the lowest mineral concentration values were found in fruit from the upper segment of branches with double girdling (TR3–G + G). The fruit from these branches had 38% lower Ca and 30% Mg contents than those from the lower segment. Additionally, in the same treatment, the fruit from the upper segment showed 62% higher N:Ca and 42% higher K:Ca stoichiometric ratios than those of fruit from the lower segment (Table 3).

The averaged Ca concentration in fruits (mg 100 g^−1^) was significantly related to the mean gs value of each treatment (Figure 4). The higher the gs, the lower the Ca concentration in the fruit, indicating a negative relationship between both parameters. Additionally, positive relationships between the gs and the stoichiometric ratios K:Ca and N:Ca were observed (Figure 4). The lowest K:Ca and N:Ca ratios were observed in fruit from branches with the lowest gs values.

The average Ca concentration in fruits was positively (*p* < 0.05) related to fruit firmness at harvest and negatively related (*p* < 0.05) to the total LA:F ratio (Table 4). The higher the total LA, the lower the Ca concentration in fruit. The lower the Ca concentration in fruit, the lower the fruit’s firmness. In contrast, the stoichiometric mineral ratios N:Ca and K:Ca were positively (*p* < 0.05) related to the LA:F ratio but negatively related to fruit firmness (*p* < 0.05). The stoichiometric ratio Mg:Ca was not significantly related to either LA:F ratios or fruit firmness.

## 3. Discussion

The phloem is the pathway for CHO transport from leaves and storage reserves to developing aerial sinks (i.e., flowers, fruits, and shoots) in sweet cherry [39]. Branches with phloem girdling at their base applied eight DAFB did not register differences in yield compared to branches with intact vascular bundles. However, unlike the branches with double girdling and branches without vascular restriction, branches with girdling at their base only had the lowest TSS and the highest firmness. These branches developed lower total LA as a consequence of fewer CSS and smaller spur leaves. High–quality fruit in sweet cherry has been associated with higher LA:F ratios, which increase fruit weight and promote higher TSS and skin–color development [40,45,46,47,48]. It is likely that, during fruit development, individual branches of the combination ‘Lapins’/Colt, trained as Kym Green Bush, either require import CHO from leaves on neighbor branches or translocation of the CHO reserves from roots and woody organs, since vascular restrictions at the base of the branch after full bloom, affected fruit quality.

In contrast, branches with double girdling did show reduced yield on the older (lower) fruiting spur segment. In these branches, the 3 and 4–year–old fruiting wood was isolated, by using girdling at their base and on the change of year between the 2–and 3–year–old wood, inducing a temporary basipetal (i.e., CHO from the root, trunk, and other vertical branches close by) and acropetal (i.e., CHO from leaves on fruiting spurs on the 1 and 2–year–old wood) source limitation during the period of fruit development. As consequence, fruit on the lower segment of these branches received CHO from leaves on the 3–and 4–year–old spurs and CSS that developed below the apical girdling only (Figure 5). It is highly possible that the reduced fruit retention detected on the bottom segment of the branch occurred because photoassimilates from leaves on the isolated fruiting spur segment were not enough sources to support reproductive growth. Interestingly, although branches with double girdling developed a higher number of lateral CSS, below the girdling between the 2–and 3–year–old wood, the additional LA per branch did not compensate the CHO´s acropetal and basipetal restriction due to the phloem cuts (Figure 5). In fact, the CSS that developed below the girdling between the 3–and 4–year–old wood might have competed with fruits on older fruiting spurs since early Stage I, inducing lower fruit retention. In sweet cherry, CSS become net exporters 25 DAFB [37,39] and, previously, competition for storage reserves among growing sinks occurs [37,40]. After Stage I of fruit development, the fruiting spur leaves are important sources of CHO followed by the leaves on non–fruiting spurs and CSS [37,44].

Girdling advanced fruit maturity without reducing fruit quality in peaches [41,42], plums [43], and sweet cherries [49]. On the other hand, increases in the LA per fruit accelerated fruit ripening in the sweet cherry combination ‘Lapins’/Gisela 5 [47]. Fruit from girdled branches (10 days before full bloom) had higher weight and TSS than fruit from branches without girding in the combinations ‘Lapins’/Maxma14 and ‘Skeena’/Maxma14 [50]. Despite that, fruit on the lower segment of branches with double girdling was harvested five days later than branches without vascular restrictions and had significantly lower TSS and TA. In these branches, a source limitation because of competition between lateral CSS and young fruit for CHO, supplied by mature spur leaves on the isolated 3–and 4–year–old wood, would have reduced the TSS and skin–color in fruits, delaying harvest in the lower segment. Increased competition for photoassimilates between CSS and fruit might have reduced fruit quality, confirming that when source photoassimilates are insufficient to supply the various sinks, fruit quality, and vegetative vigor will be less than optimal [39]. On the contrary, fruit on the upper segment of the same branches was harvested five days earlier, compared with branches without vascular restrictions. Besides, the fruit from the upper segment was larger and registered higher TSS and TA than the fruit from branches without girdling. It seems that the girdling on the change of year, between the 2–and 3–year–old fruiting spur wood (Figure 5 and Table 2), restricted the basipetal translocation of photoassimilates from the upper segment since early Stage I [44,51], promoting a higher accumulation of CHO in fruit of the 1–and 2–year old wood, which in turn, improved fruit quality (Figure 2).

Several studies in sweet cherry have concluded that firmness is positively related to higher LA:F ratios [40,45,46,52], which does not agree with our results. The Ca concentration was reduced in fruit harvested from the upper segment of double–girdled branches only, where additional lateral CSS developed below the girdling (between the 2–and 3–year–old wood). However, fruit from this segment was the largest (29.5 mm), the sweetest (18.6%), the most acidic (1.0%), and had the higher dry matter concentration (21.2%), despite firmness (67.5 durofel unit) was significantly lower. We hypothesize that the lower fruit firmness observed on the upper segment of double–-girdled branches might be a consequence of changes in the mineral composition of fruit, particularly associated with the Ca concentration. In sweet cherry, the highest Ca concentration in fruit is observed in the early stages of development, after which, a continuous decrease occurs [1,53,54]. Calcium dilution by cell expansion and reduced Ca uptake by the xylem stream have been reported [30]. The highest the transpiration rate of the organ (i.e., high surface volume of the fruit in relation to mass), the highest the Ca transport to that organ [13]. The fruit located in the upper segment of branches with double girdling had high dry matter concentration and TSS, but lower Ca concentration than the other treatments. It is possible that, in the lower segment of these branches, the higher number of CSS below the apical girdling (Table 1 and Figure 5) might have caused preferential distribution of Ca towards actively growing shoots, which have higher transpiration rates than the fruit located above the apical girdling [17]. We propose that the double girdling in vertical branches did not affect fruit Ca concentration per se but promoted the growth of CSS below the apical girdling. The higher growth rate and transpiration of the CSS compared to the fruit on the upper fruiting segment might have influenced Ca distribution and, consequently, decreased the Ca concentration of the fruit in the upper segment (Figure 6). The unbalance in mineral nutrition occurred also at the level of K and N, promoting the highest K:Ca and N:Ca ratios, in the fruit of the upper section.

Vertical branches with cutting 50% of the xylem cross–section area only and those with restriction to the xylem and girdling at their base did not register decreases in Ca concentration or Ca content in fruit compared with the branches without xylem limitations. It is well known that the accumulation of Ca in aerial sinks, such as fruit, depends on its supply by the xylem stream [21] and it is affected by transpiration rates [55]. The higher the fruit´s transpiration rate, the more Ca will be transported to that organ [12,13]. Deficiency symptoms occur more frequently in developing tissues, such as young fruits, due to the poor mobility of Ca nutrients within the plant. In spite of this, the xylem cuts did not negatively affect acropetal Ca translocation to fruit in this experiment.

On the other hand, fruit from the branches with a cross–sectional cut to the xylem and girdling at their base had the lowest dry matter content and the smallest fruit size (Figure 2). Dry matter distribution is the end result of a coordinated set of transport and metabolic processes governing the flux and distribution of CHO from source organs to sink organs [38]. Therefore, in these branches, although CHO availability was restricted due to girdling, negatively affecting fruit size, Ca movement continued to be driven by transpiration, despite xylem restriction during fruit development.

We observed negative relationships between fruit Ca concentration and the midday gs values (Figure 4) within the same fruiting segment, as well as between the Ca concentration in the fruit and the LA:F ratio (Table 4). In apple, increments in LA or high LA:F ratios, negatively affect Ca translocation to the fruit. A higher amount of Ca is distributed to leaves, causing a nutrient imbalance between leaves and fruits, modifying N:Ca and K:Ca ratios [56]. Moreover, apple fruit borne on spurs with larger leaves had higher Ca concentrations than fruit borne on spurs with smaller leaves [14] and reductions in shoot growth, increased Ca content in fruit [57]. Although sweet cherry does not present bourse shoots that can either compete for or promote Ca distribution towards the fruit, CSS developed near the fruit might influence Ca distribution to them prior to harvest.

Some authors indicate that Ca concentration and firmness in sweet cherries are positively related [58,59,60] and physiological disorders, such as bitter pit in apples, have been directly associated with N:Ca, K:Ca, and Mg:Ca ratios [61,62]. Low Ca concentration (7.7 mg 100 g^−1^) was associated with low fruit firmness in ‘Santina’ sweet cherry grown under plastic covers [63]. We found that fruit firmness was significantly and positively related to fruit Ca concentration, but negatively related to the N:Ca and K:Ca ratios (Table 4) [26].

The phloem and xylem restrictions imposed on the branches and their indirect effect on the final fruit set influenced nutrient concentration in fruit. Fruit from the upper segment of branches with double girdling registered lower Ca concentrations and lower firmness than the fruits from the upper segment of branches without vascular intervention. In all treatments, Mg concentrations (8 to 13 mg 100 g^−1^) ranged among the values described by Serradilla et al. (2017) [64], with the exception of fruits from the upper segment of branches with double girdling that showed the lowest values. Low Ca concentrations may explain the reduction in Mg in the same fruits, since the Ca signaling pathway regulates Mg homeostasis [9]. On the other hand, Neilsen et al. (2007) [65], reported that crop load level affected the fruit´s mineral composition, with fruit thinning promoting higher contents of N and K in the remaining fruits of the combination ‘Lapins’/Gisela 5. Despite vascular restrictions, N concentrations in fruits were within or above the upper limit of the range (133 to 188 mg 100 g^−1^) reported by Neilsen et al. (2007) [65] for the sweet cherry cv. ‘Lapins’. Although, our results were not similar to those reported by Michailidis et al. (2020) [50], who observed that from 2–year–old girdled branches (10 d before full bloom) of the cv. ‘Lapins’, the N content in fruit increased.

According to the above, vascular restrictions in vertical branches of the combination ‘Lapins’/Colt, imposed early in the season, induced changes in the CHO distribution, LA development, and fruit mineral composition, which in turn, impacted fruit quality and yield. We propose that the lowest Ca concentration in sweet cherries grown in vertical branches might be a result of the highest transpiration rates of lateral CSS induced by the second (upper) girdling and developed during the growing season. Additional efforts are required to integrate cultural practices that promote CHO and Ca distribution in sweet cherry, considering different sweet cherry combinations, crop load levels, and novel training systems.

## 4. Materials and Methods

### 4.1. Plant Material

The study was conducted during the 2019–20 growing season in a five–year old commercial cherry orchard with the highly–productive, self–fertile combination ‘Lapins’/Colt, located in San Francisco de Mostazal, Sixth Region, Chile (34° S, 70°41′ W). The irrigation system consisted of two drip irrigation lines per row and eight emitters of 2 L h^−1^ per tree. Irrigation frequency varied from one irrigation event per week early in the season (Sep) to two irrigation events per week (Oct–Nov). Each irrigation event consisted of a single daily irrigation of 14 h. Trees were trained to the Kym Green Bush (KGB) system (21 vertical branches per tree), at a tree spacing of 2.0 m (in the row) × 3.8 m (between rows), with east–west row orientation. Crop load regulation was not carried out during the experiment and trees were not protected from rain or birds. The main commercial practices included weed control and soil mineral nutrition using 10 kg N ha^−1^, 10 kg P ha^−1^, and 6 kg B ha^−1^. Two foliar sprays of Ca were applied as 40 kg CaO ha^−1^. The first at: 50% bloom and the second at 20 days after full bloom (DAFB) (i.e., fruit set). Full bloom (FB) occurred on 30 August 2019.

Before imposing the treatments, trees were characterized in terms of numbers and diameters (mm) of branches and numbers and types of spurs (i.e., fruiting or non–fruiting) per branch. The numbers of current–season shoots (CSS) per branch were recorded at harvest.

### 4.2. Experimental Design and Treatments

Ten rows of trees were selected within the orchard and, from each row, ten replications of five trees of similar vigor, number of branches, and height were used for each treatment (TR). From each individual tree, a single vertical branch was selected. This branch and this tree formed one experimental unit.

Vascular restrictions were of two sorts. (1) Girdling (G): here the phloem supply was stopped completely by girdling, the removal of a complete ring of bark, cut down to the wood with a bander knife (3/16′ (4.76 mm); Zenport GK02, Taiwan) and (2) transverse incision (S): here the branch was sawn through to the halfway point with a hand saw (Redline 18H108E2, Taiwan), reducing the xylem supply to the distal parts of the shoot, by severing ~50% of xylem cross–section area at one point.

Five treatments (TR1, TR2… TR5) were imposed that combined these vascular restrictions in the following pattern. TR1–CTL: a control branch with neither G nor S at its base, TR2–G: a branch with G at the base (5 cm from the trunk’s crown), TR3–G + G: a branch with G at the base (5 cm from the trunk’s crown) and G further up, at the change of year between the second and the third years of growth, TR4–S: a branch with S at the base (5 cm from the trunk’s crown) and TR5–S + G: a branch with S and G at the base (S 5 cm from the trunk’s crown and G 1 cm above S). Treatments were imposed on eight DAFB on 7 September 2019 (Day of year, DOY 250). Tools were disinfected with 70% alcohol prior to each use and a fungicide solution (Captan 80 WP 0.8 g L^−1^ plus Benomyl 50 PM 1 g L^−1^) was applied to the treatment area with a manual sprayer. Moreover, the G and S treatment areas were inspected five times during the season to eliminate recovery growth of the vascular tissues (see Quentin et al., (2013) [49]). In no cases did these treatments result in the death of a treated branch during the experimental period.

Each branch was considered in two halves (segments)—the upper segment and the lower segment. The lower segment was defined as running from the base of the branch (from the crow) to the change of year, between the second and the third year of growth, and the upper segment, from the change of year between the second and the third year of growth, including the shoot tip (Figure 5).

### 4.3. Field Measurements

At harvest, the branch yield and the number of leaves and leaf area (LI–3100, LI–COR Inc., Lincoln, NE, USA) of four spurs and five terminal shoots from five branches per treatment were recorded. The yield (kg) of each branch was recorded with a digital scale (Sudstar, China). In addition, the number of spurs, CSS, and fruits per branch were used to estimate the total LA of individual branches, the tree yield, and the LA:F ratio (cm^2^ fruit^−1^), according to the methodology used by Whiting and Lang, (2004) and Blanco et al. (2019) [40,66].

### 4.4. Environmental and Physiological Measurements

Climatic data such as air temperature (°C), relative humidity (%) and vapor pressure deficit (VPD, kPa) were recorded by a meteorological station located in the orchard (iMetos 3.3 Pessl Instruments GmbH, Weiz, Austria). Stomatal conductance (gs, mmol m^–2^ s^−1^), was measured weekly at noon during the Stage III of fruit development (i.e., 50–85 DAFB) with a portable porometer (SC–1, Decagon Devices Inc., Pullman, WA, USA) in three leaves per tree, from spurs with the same orientation and solar exposure, and three trees per treatment [66]. On branches from TR3–G + G, with double girdling, two segments were distinguished for measurements—the lower segment and the upper segment.

### 4.5. Fruit Quality at Harvest

Commercial harvest (i.e., between 87 and 92 DAFB) was determined using fruit color as an indicator and based on a fruit skin–color table (3.5 color cherry color chart, Pontificia Universidad Católica de Chile).

Fruit quality parameters were measured immediately after harvest, using six replications of 30 fruits per treatment. Fruit diameter (mm) was measured with a digital caliper (Hubermann, Santiago, Chile), the fresh unitary weight (g) with a digital scale (Pocket Scale, HM–Series, Wenzhou, China) and the TSS (%) through an optical temperature compensated refractometer (Veto, Santiago, Chile). The TA (g L^−1^) was determined with a pH meter (Veto, Santiago, Chile) using 5 mL of juice plus 45 mL of distilled water and 0.1 N NaOH until reaching a pH of 8.1–8.2. Fruit firmness was measured over the range 0 (soft) to 100 (firm) using a durometer (type A, Durofel Agrotechnologie, Tarascon, France) with a 2.5 mm tip.

Fruit dry matter (%) was calculated as the percentage of the dry weight of the sample (oven–dried at 60 °C for 48 h until constant weight) relative to its fresh weight. From these measurements and yield per each branch, the total fruit dry weight by individual branches was calculated.

### 4.6. Fruit Mineral Analysis

Mineral composition at harvest was determined using 500 g of fruit per replicate as samples, six samples per treatment (three to each of the two segments). Mineral concentrations of Ca, K, and Mg were analyzed by dry combustion until components were converted to ash, according to the methodology of Ryan et al. (2001) [67]. The ashed tissue samples were dissolved in HCl (2 M), and concentrations were determined by Inductively Coupled Plasma–Optical Emission Spectroscopy (ICP–OES) (Agilent 720 ES axial–Varian, Mulgrave, Victoria, Australia). The N concentrations were determined with a LECO CNS–2000 Macro Elemental Analyzer (Leco, St. Joseph, MI, USA). From these measurements, fruit mineral concentration (mg·100 g^−1^ of fresh fruit), fruit mineral content (mg·fruit^−1^) and the stoichiometric ratios N:Ca, K:Ca and Mg:Ca were calculated.

### 4.7. Statistical Analyses

The results were analyzed using statistical analysis, ANOVA (Analysis of variance; *p* = 0.05) to determine differences between treatments, followed by a Tukey test. The effects of the upper and lower branch segments on yield and fruit quality (unitary weight, dry matter, diameter, TSS, TA, and firmness) were evaluated by ANOVA using a factorial treatment arrangement. The relationships among variables were assessed by comparing Pearson’s correlation coefficients. Statistical analyses were carried out using the R studio package (Rstudio Inc., Boston, MA, USA).

## 5. Conclusions

Sweet cherry branches with xylem restrictions alone, or with xylem restrictions combined with phloem restrictions, did not induce changes in Ca concentration in fruit. Calcium concentration was reduced in fruit harvested from the upper segment of double–girdled branches only, where additional lateral CSS developed below the girdling between the 2–and 3–year–old wood. However, fruit from this segment was the largest (29.5 mm), the sweetest (18.6%), the most acid (1.0%), and had the higher dry matter concentration (21.2%), although firmness (67.5 durofel unit) was significantly lower. The Ca concentration in fruit was higher (13.8 mg 100 g^−1^) in the lower segment of the branch, while in the upper segment was lower (6.6 mg 100 g^−1^) (Table 3). We hypothesize that in these branches, CSS growth competed with developing fruit for Ca, reducing its concentration in fruit, which in turn, negatively affected firmness. Calcium concentration in fruit was negatively related to the stomatal conductance of the leaves from the same segment of the branch. Moreover, fruit firmness was positively related to fruit Ca concentration and negatively related to the ratios of K:Ca and N:Ca.

Further research is required to continue elucidating the role played by actively growing CSS in modifying Ca concentrations in fruit and its implications in determining final fruit quality and storage potential. The information provided from such research will allow the development of new agronomic practices aimed at increasing fruit Ca concentration and firmness.

## Figures and Tables

**Figure 1 plants-12-01922-f001:**
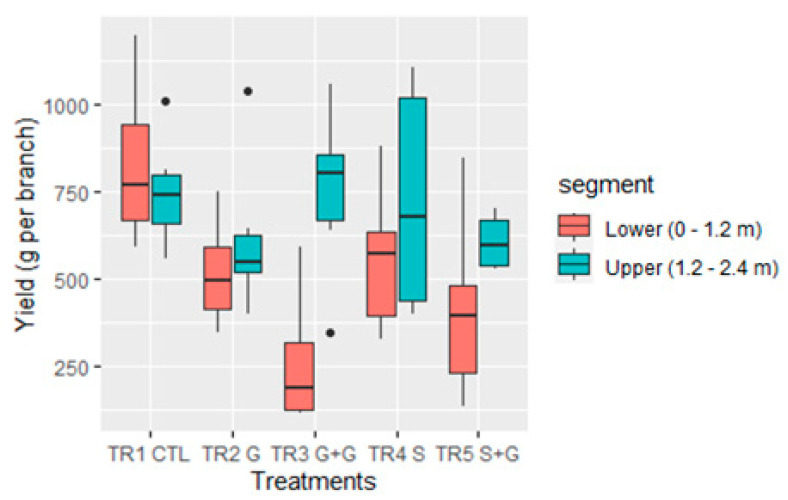
Fruit yield (g) of individual vertical branches with vascular restrictions, in the scion rootstock combination ‘Lapins’/Colt, trained as Kym Green Bush (*n* = 10). TR1–CTL: a control branch with neither girdling (G) nor transverse incision cutting 50% of xylem cross–section area (S), TR2–G: a branch with G at its base, TR3–G + G: a branch with G at its base and G further up at the change of year between the second and the third years of growth, TR4–S: a branch with S at its base and TR5–S + G: a branch with S and G at its base. Lower segment from the crown to 1.2 m height on the branch. Upper segment from 1.2 m to 2.4 height on the branch.

**Figure 2 plants-12-01922-f002:**
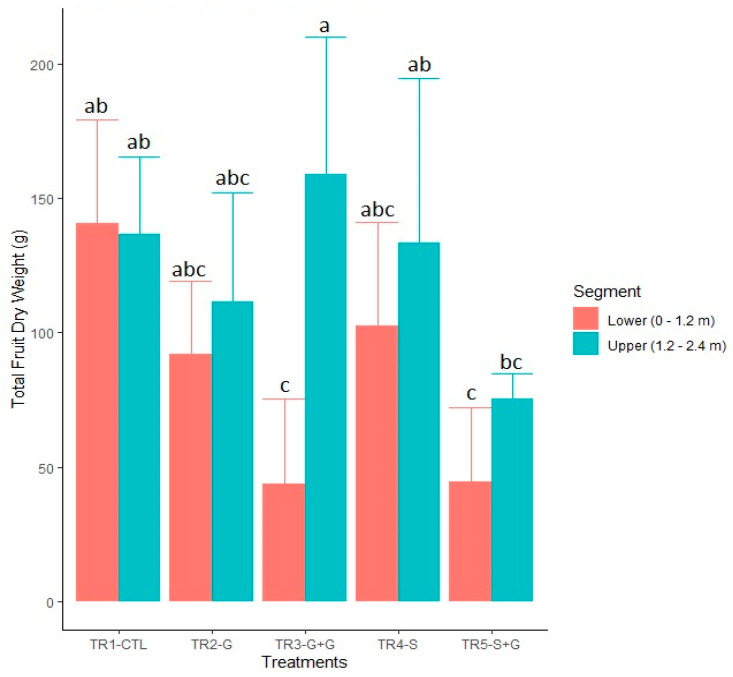
Total fruit dry weight (g) produced by individual vertical sweet cherry branches with vascular restrictions, in the scion rootstock combination ‘Lapins’/Colt, trained as Kym Green Bush. Each column is expressed as mean value and standard deviation (*n* = 6). Different letters within the same column indicate significant differences (*p* ≤ 0.05) among treatments by Tukey’s test. TR1–CTL: a control branch with neither girdling (G) nor transverse incision cutting 50% of xylem cross–section area (S), TR2–G: a branch with G at its base, TR3–G + G: a branch with G at its base and G further up at the change of year between the second and the third years of growth, TR4–S: a branch with S at its base and TR5–S + G: a branch with S and G at its base. Lower segment from the crown to 1.2 m height on the branch. Upper segment from 1.2 m to 2.4 height on the branch.

**Figure 3 plants-12-01922-f003:**
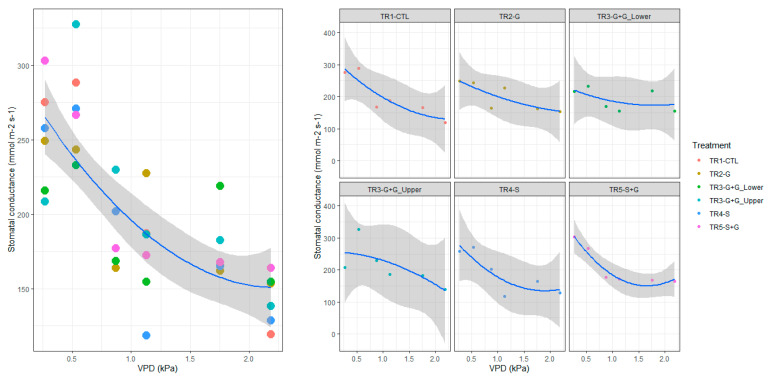
Relationship of stomatal conductance (gs) with vapor pressure deficit (VPD) of leaves from vertical sweet cherry branches with vascular restrictions, in the scion rootstock combination ‘Lapins’/Colt, trained as Kym Green Bush (*n* = 6). TR1–CTL: a control branch with neither girdling (G) nor transverse incision cutting 50% of xylem cross–section area (S), TR2–G: a branch with G at its base, TR3–G + G: a branch with G at its base and G further up at the change of year between the second and the third years of growth, TR4–S: a branch with S at its base and TR5–S + G: a branch with S and G at its base. Lower segment from the crown to 1.2 m height on the branch. Upper segment from 1.2 m to 2.4 height on the branch.

**Figure 4 plants-12-01922-f004:**
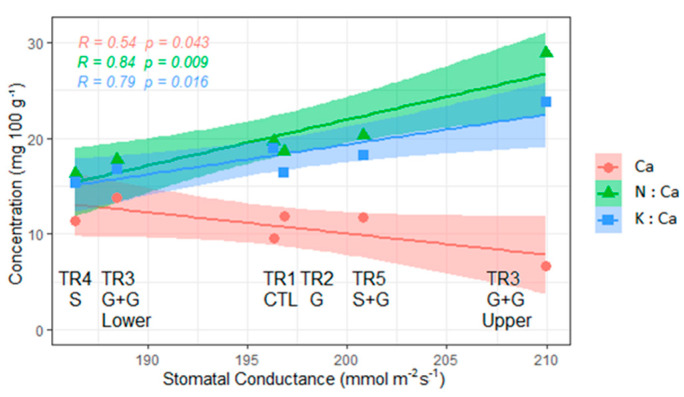
Relationship between mean stomatal conductance (gs), Ca concentration in fruit (mg 100 g^−1^) and, stoichiometric ratios (N:Ca and K:Ca) during Stage III of fruit development in sweet cherry from vertical branches with vascular restrictions, in the scion rootstock combination ‘Lapins’/Colt, trained as Kym Green Bush. TR1–CTL: a control branch with neither girdling (G) nor transverse incision cutting 50% of xylem cross–section area (S), TR2–G: a branch with G at its base, TR3–G + G: a branch with G at its base and G further up at the change of year between the second and the third years of growth, TR4–S: a branch with S at its base and TR5–S + G: a branch with S and G at its base. Lower segment from the crown to 1.2 m height on the branch. Upper segment from 1.2 m to 2.4 height on the branch.

**Figure 5 plants-12-01922-f005:**
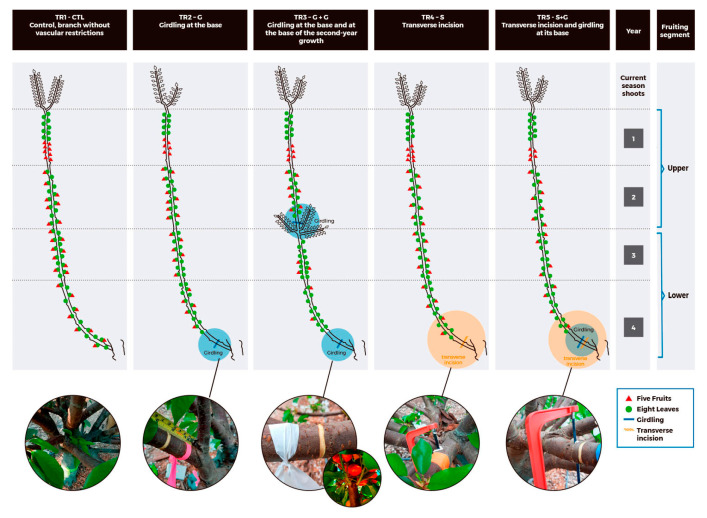
Diagram of vascular restriction treatments, TR1–CTL: a control branch with neither girdling (G) nor transverse incision (S), TR2–G: a branch with G at its base, TR3–G + G: a branch with G at its base and G further up at the change of year between the second and the third years of growth, TR4–S: a branch with S at its base and TR5–S + G: a branch with S and G at its base. Year 1: non–fruiting spurs and Year 2, 3, and 4: fruiting spurs.

**Figure 6 plants-12-01922-f006:**
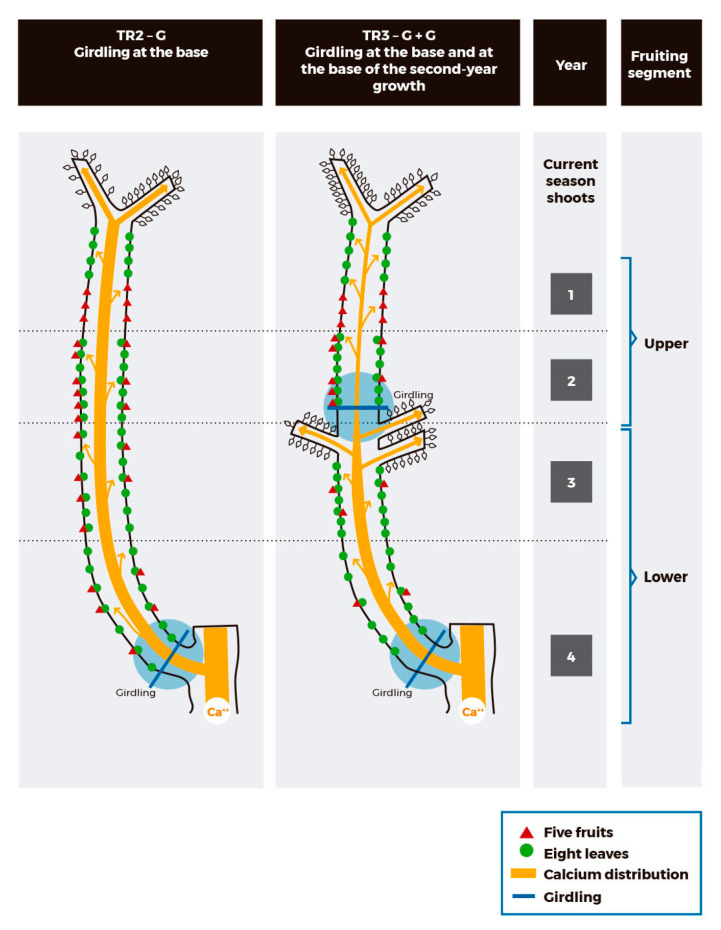
Proposed outline of Ca distribution in vertical sweet cherry girdled branches in KGB. Treatment included: TR2–G branch (with girdling at its base) and TR3–G + G branch (with double girdling, at its base and further up at the change of year between the second and the third years of growth).

**Table 1 plants-12-01922-t001:** Leaf area at harvest (current season shoots and spurs), number of fruits and leaf area to fruit ratio from vertical sweet cherry branches with vascular restrictions, in the scion rootstock combination ‘Lapins’/Colt, trained as Kym Green Bush.

	Current Season Shoots		Spurs		Branch	Fruit	LA:F
	Number	Leaf Area	Number	Leaf Area	Total Leaf Area	Number	Total Leaf Area/n° Fruits
Treatments		(cm^2^)		(cm^2^)	(cm^2^)		(cm^2^·F^−1^)
TR1–CTL	2.0 b	1178	48.5	230 ab	13,501 ab	146 a	104 b
TR2–G	1.8 b	1080	48.5	181 b	10,676 b	108 abc	98.6 b
TR3–G + G	5.4 a	522	46.3	275 a	15,560 a	90.5 c	180 a
TR4–S	2.0 b	914	48.7	201 b	11,438 b	141 ab	85.7 b
TR5–S + G	2.2 b	1203	46.7	208 b	12,315 ab	98.0 bc	134 ab
*p*-value	<0.001	0.197	0.983	0.001	0.016	0.005	0.008

Different letters within the same column indicate significant differences (*p* ≤ 0.05) among treatments by Tukey’s test (*n* = 10). TR1–CTL: a control branch with neither girdling (G) nor transverse incision cutting 50% of xylem cross–section area (S), TR2–G: a branch with G at its base, TR3–G + G: a branch with G at its base and G further up at the change of year between the second and the third years of growth, TR4–S: a branch with S at its base and TR5–S + G: a branch with S and G at its base. Lower segment from the crown to 1.2 m height on the branch. Upper segment from 1.2 m to 2.4 height on the branch.

**Table 2 plants-12-01922-t002:** Quality parameters of fruit at harvest from vertical sweet cherry branches with vascular restrictions, in the scion rootstock combination ‘Lapins’/Colt, trained as Kym Green Bush.

**Upper Segment**	**Unitary Fresh Weight**	**Dry Weight**	**Diameter**	**Firmness**	**TSS**	**TA**
**Treatments**	**(g)**	**(%)**	**(mm)**	**(Durofel Unit 0–100)**	**(%)**	**(%)**
TR1–CTL	9.3 ^b^	18.3 ^a^	28.9 ^ab^	71.7 ^ab^	17.3 ^ab^	0.97 ^ab^
TR2–G	8.1 ^b^	18.2 ^a^	28.4 ^bc^	73.2 ^a^	15.6 ^b^	0.91 ^abc^
TR3–G + G	11.7 ^a^	21.2 ^a^	29.5 ^a^	67.5 ^b^	18.6 ^a^	1.0 ^a^
TR4–S	9.3 ^b^	18.4 ^a^	28.0 ^c^	70.3 ^ab^	16.4 ^b^	0.84 ^c^
TR5–S + G	9.8 ^b^	12.5 ^b^	27.0 ^d^	69.8 ^ab^	16.5 ^b^	0.87 ^bc^
*p*–value	0.0001	<0.0001	<0.0001	0.043	0.001	0.002
**Lower Segment**	**Unitary Fresh Weight**	**Dry Weight**	**Diameter**	**Firmness**	**TSS**	**TA**
**Treatments**	**(g)**	**(%)**	**(mm)**	**(Durofel Unit 0–100)**	**(%)**	**(%)**
TR1–CTL	9.8	17.0 ^a^	28.4 ^a^	72.3	15.7	0.93
TR2–G	9.8	17.9 ^a^	28.0 ^ab^	73.8	14.5	0.88
TR3–G + G	10.3	17.1 ^a^	27.4 ^b^	72.4	15.5	0.86
TR4–S	10.2	18.5 ^a^	28.2 ^a^	70.0	15.4	0.81
TR5–S + G	10.3	10.9 ^b^	27.3 ^b^	69.0	15.6	0.89
*p*–value	0.938	<0.0001	<0.0001	0.059	0.607	0.370

Different letters within the same column indicate significant differences (*p* ≤ 0.05) among treatments by Tukey’s test (*n* = 6). TR1–CTL: a control branch with neither girdling (G) nor transverse incision cutting 50% of xylem cross–section area (S), TR2–G: a branch with G at its base, TR3–G + G: a branch with G at its base and G further up at the change of year between the second and the third years of growth, TR4–S: a branch with S at its base and TR5–S + G: a branch with S and G at its base. Lower segment from the crown to 1.2 m height on the branch. Upper segment from 1.2 m to 2.4 height on the branch.

**Table 3 plants-12-01922-t003:** Total calcium (Ca), nitrogen (N), potassium (K) and magnesium (Mg) content (mg fruit^−1^) and concentration (mg 100 g^−1^), and stoichiometric ratios (N:Ca, K:Ca and Mg:Ca) in fruit at harvest from vertical sweet cherry branches with vascular restrictions, in the scion rootstock combination ‘Lapins’/Colt, trained as Kym Green Bush.

**Upper** **Segment**	**Ca**		**N**		**K**		**Mg**		**N:Ca**	**K:Ca**	**Mg:Ca**
**Treatments**	**mg fruit^−^** ** ^1^ **	**mg 100 g^−1^**	**mg fruit^−^** ** ^1^ **	**mg 100 g^−1^**	**mg fruit^−^** ** ^1^ **	**mg 100 g^−1^**	**mg fruit^−^** ** ^1^ **	**mg 100 g^−1^**			
TR1–CTL	1.1	9.9	20.3	181	18.3	163	1.2	11.0 a	18.4 ab	16.5 b	1.1
TR2–G	1.2	11.9	23.3	232	19.7	197	1.3	13.0 a	20.1 ab	17.0 b	1.1
TR3–G + G	0.9	6.6	24.6	191	19.9	156	0.9	6.9 b	28.8 a	23.7 a	1.1
TR4–S	1.2	12.0	17.0	179	16.2	171	1.1	11.6 a	15.6 b	14.9 b	1.0
TR5–S + G	1.2	12.8	20.8	199	18.3	199	1.1	12.3 a	18.0 ab	15.9 b	1.0
*p*–value	0.561	0.072	0.306	0.203	0.422	0.073	0.155	0.005	0.040	0.027	0.335
**Lower** **Segment**	**Ca**		**N**		**K**		**Mg**		**N:Ca**	**K:Ca**	**Mg:Ca**
**Treatments**	**mg fruit^−^** ** ^1^ **	**mg** **100 g^−1^**	**mg fruit** ** ^−1^ **	**mg** **100 g^−1^**	**mg fruit** ** ^−1^ **	**mg** **100 g^−1^**	**mg fruit** ** ^−1^ **	**mg** **100 g^−1^**			
TR1–CTL	1.0	9.2	19.2	183	19.4	186	1.0	9.7	21.0	21.4	1.1
TR2–G	1.2	11.9	20.5	203	19.1	189	1.2	11.6	17.0	15.9	1.0
TR3–G + G	1.4	13.8	22.3	221	21.2	209	1.3	12.4	17.8	16.7	1.0
TR4–S	1.0	10.6	17.7	179	16.5	168	1.1	10.7	17.0	15.9	1.0
TR5–S + G	1.1	10.7	23.9	225	21.7	207	1.1	10.6	22.6	20.6	1.0
*p*–value	0.607	0.471	0.645	0.299	0.293	0.452	0.239	0.217	0.658	0.524	0.839

Different letters within the same column indicate significant differences (*p* ≤ 0.05) among treatments by Tukey’s test (*n* = 3). TR1–CTL: a control branch with neither girdling (G) nor transverse incision cutting 50% of xylem cross–section area (S), TR2–G: a branch with G at its base, TR3–G + G: a branch with G at its base and G further up at the change of year between the second and the third years of growth, TR4–S: a branch with S at its base and TR5–S + G: a branch with S and G at its base. Lower segment from the crown to 1.2 m height on the branch. Upper segment from 1.2 m to 2.4 height on the branch.

**Table 4 plants-12-01922-t004:** Pearson’s correlation coefficient between the leaf area to fruit (LA:F) ratio, Ca concentration (Ca), stoichiometric ratios (N:Ca, K:Ca and Mg:Ca), and fruit firmness in vertical branches of the sweet cherry fruit combination ‘Lapins’/Colt, trained as Kym Green Bush, at harvest.

	LA:F (cm^2^·F^−1^)	Firmness (Durofel Unit 0–100)
Ca (mg 100 g^−1^)	* −0.68	* 0.43
N:Ca	* 0.70	* −0.29
K:Ca	* 0.82	* −0.39
Mg:Ca	0.02	−0.19

Values marked with ***** are significant at *p* < 0.05.

## Data Availability

Raw data is available upon request from the corresponding author.

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
