# Peer review of "Study of Mineral Composition and Quality of Fruit Using Vascular Restrictions in Branches of Sweet Cherry"

_plants, 2023, doi:10.3390/plants12101922_

Round 1
Reviewer 1 Report (New Reviewer)
The paper reports on an interesting investigation. The results and conclusions give an insight into the sink - source balance of sweet cherry leaves, shoots and fruits, influencing the fruit quality attributes. The results and conclusions may give useful information to those horticultural practices like trunk incision or girdling, which are applied on vigorous trees to overcome the negative effect of a severe flower frost damage. However, further research is needed to illuminate the role of leaves on fruiting branches and current season shoots to develop proper practice for Kym Green Bush trees on vigorous rootstocks. I would suggest to emphasize in the introduction (last paragraph) that the possible practices target to improve the fruit and storage quality on Kym Green Busht training system.
Author Response
The paper reports on an interesting investigation. The results and conclusions give an insight into the sink - source balance of sweet cherry leaves, shoots and fruits, influencing the fruit quality attributes. The results and conclusions may give useful information to those horticultural practices like trunk incision or girdling, which are applied on vigorous trees to overcome the negative effect of a severe flower frost damage. However, further research is needed to illuminate the role of leaves on fruiting branches and current season shoots to develop proper practice for Kym Green Bush trees on vigorous rootstocks. I would suggest to emphasize in the introduction (last paragraph) that the possible practices target to improve the fruit and storage quality on Kym Green Busht training system.
A1. We considered your comments and included changed in the manuscript. Changes are highlighted in the text line 102 – 106.
Please see the attachment.

Reviewer 2 Report (Previous Reviewer 3)
- The authors have addressed the reviewer comments and suggestions.
(The type of statistical analysis carried out could be defined in the text not only in the description at the bottom of the tables).
Author Response
- The type of statistical analysis carried out could be defined in the text not only in the description at the bottom of the tables.
A1. Please see line del review manuscript 553 – 560 in the section M&M as follows:
“The results were analyzed using statistical analysis, ANOVA (Analysis of variance; p = 0.05) to determine differences between treatments, followed by a Tukey test. The effects of the upper and lower branch segments on yield and fruit quality (unitary weight, dry matter, diameter, TSS, TA and firmness) were evaluated by ANOVA using a factorial treatment arrangement. The relationships among variables were assessed by comparing the Pearson's correlation coefficients. Statistical analyses were carried out using R studio package (Rstudio Inc., Boston, MA, USA).”
Please see attachment to check M&M.

Reviewer 3 Report (New Reviewer)
Dear Authors,
Manuscript is interesting and provides important data of xylem and Phloem flow of Ca and minerals.
But some improvements in methodology must be improved.
Two main complaints goes to
1. Durofel unit. It is not precise unit. Usage of SI unit (N) should be better and comparable.
2. Mineral content per fruit (mg fruit-1) is unnecessary data that is not comparable to anything else. Each fruit has different weight thus mineral content per fruit doesn't mean nothing.
Also my suggestion is to move chapter Materials & Methods after Introduction and before Results. When reader is familiar how experiment and analysis were conducted and results will be clear for him.
Specific comments are given in manuscript.
Regards

Author Response
- Durofel unit. It is not precise unit. Usage of SI unit (N) should be better and comparable.
A1. We would prefer to report the measurements of firmness directly as was obtained by the Durofel.
Durofel was used since it is a specific durometer (Type A) developed for measuring the hardness of sweet cherry tissue. The Durofel is recognized as instrument, among others, to characterize the sweet cherry firmness in a scale from 0 to 100. The durometer value can be transformed to the Young´s modulus by Eq. 1. However, the value of Young's modulus (MPa) or modulus of elasticity are a complex response of the tensile strength of the tissue that is better to obtain directly from a compression test and this was not the purpose of this work.
Eq 1=
- Durofel unit. It is not precise unit. Usage of SI unit (N) should be better and comparable.
A1.
We would prefer to report the measurements of firmness directly as was obtained by the Durofel.
Durofel was used since it is a specific durometer (Type A) developed for measuring the hardness of sweet cherry tissue. The Durofel is recognized as instrument, among others, to characterize the sweet cherry firmness in a scale from 0 to 100. The durometer value can be transformed to the Young´s modulus by Eq. 1 (see Eq 1 at the end of attached manuscript). However, the value of Young's modulus (MPa) or modulus of elasticity are a complex response of the tensile strength of the tissue that is better to obtain directly from a compression test and this was not the purpose of this work.
** For further information see:
Clayton, M., Biasi, B. and Mitcham, B. 1998. Devices for measuring firmness of cherries. Perishables Hand-ling Quarterly 95, 2–4. Available at: https://ucanr.edu/datastoreFiles/234-52.pdf
Mix A. W. and Giacomin A. J. 2011. Standardized Polymer Durometry, Journal of Testing and Evaluation,39(4), pp. 1–10. Doi:10.1520/JTE103205
- Mineral content per fruit (mg fruit-1) is unnecessary data that is not comparable to anything else. Each fruit has different weight thus mineral content per fruit doesn't mean nothing.
A2. We appreciate the reviewer's suggestion, however, we would like to keep the data since the Ca content and concentration have different physiological and nutritional interpretations in the case of sweet cherry. Please see as reference Matteo et al., 2022 (section 2.5 M&M and Fig 1 in results).
Matteo, M.; Zoffoli, J.P.; Ayala, M. Calcium Sprays and Crop Load Reduction Increase Fruit Quality and Postharvest Storage in Sweet Cherry (Prunus Avium L.). Agronomy 2022, 12, 829, doi:10.3390/agronomy12040829.
- Also my suggestion is to move chapter Materials & Methodsafter Introductionand before Results. When reader is familiar how experiment and analysis were conducted and results will be clear for him.
A3. We agree with your comment, but the journal requires to place the M&M section after the discussion. We should not change the order according to the Editor.
Specific comments are given in manuscript.
A4. We considered your comments and included changed in the manuscript, highlight in the text line 75, 93 and 456 to 458.
** For further information see:
Clayton, M., Biasi, B. and Mitcham, B. 1998. Devices for measuring firmness of cherries. Perishables Hand-ling Quarterly 95, 2–4. Available at: https://ucanr.edu/datastoreFiles/234-52.pdf
Mix A. W. and Giacomin A. J. 2011. Standardized Polymer Durometry, Journal of Testing and Evaluation,39(4), pp. 1–10. Doi:10.1520/JTE103205
- Mineral content per fruit (mg fruit-1) is unnecessary data that is not comparable to anything else. Each fruit has different weight thus mineral content per fruit doesn't mean nothing.
A2. We appreciate the reviewer's suggestion, however, we would like to keep the data since the Ca content and concentration have different physiological and nutritional interpretations in the case of sweet cherry. Please see as reference Matteo et al., 2022 (section 2.5 M&M and Fig 1 in results).
Matteo, M.; Zoffoli, J.P.; Ayala, M. Calcium Sprays and Crop Load Reduction Increase Fruit Quality and Postharvest Storage in Sweet Cherry (Prunus Avium L.). Agronomy 2022, 12, 829, doi:10.3390/agronomy12040829.
- Also my suggestion is to move chapter Materials & Methodsafter Introductionand before Results. When reader is familiar how experiment and analysis were conducted and results will be clear for him.
A3. We agree with your comment, but the journal requires to place the M&M section after the discussion. We should not change the order according to the Editor.
Specific comments are given in manuscript.
A4. We considered your comments and included changed in the manuscript, highlight in the text line 75, 93 and 459 to 461.
Please see attachment with reviewed manuscript and answers to your questions.

This manuscript is a resubmission of an earlier submission. The following is a list of the peer review reports and author responses from that submission.
Round 1
Reviewer 1 Report
The senior author has published a number of papers on cherry physiology and quality. Although her most recent publication (Matteo et al., 2022) is listed in the references, it is not cited in the body of the text. I did not make time to check on the other references, but urge her to make sure that they are indeed cited.
It is usually considered that size, firmness, sugar, and maybe color are the attributes that consumers look for in cherries. In this study, small fruit were firmer, while big fruit were sweeter. It would have been interesting to have a taste panel determine actual preferences, or at least to cite studies showing what is really important to consumers regarding cherry quality.
The VPD studies, while interesting, do not have SD or SE bars. The lines are drawn with various curves, but seem to be mostly linear, except for the summary graph. It is not clear if or how the various treatments are affected by VPD or stomatal aperture, or if that affects fruit quality attributes mentioned earlier.
The manuscript does not properly follow the Rule of the Decimal Place. If there are no numbers to the left of the decimal point, you can present up to two numbers to the right of the decimal One or two numbers to the left of the decimal allows one digit to the right of the decimal. Three or more digits to the left means no digits to the right of the decimal, and in fact no decimal is needed in that case.
Figure 1-- why is this presented as a box-plot?
Figure 2 Are these SD or SE bars? If these are Tukey mean separations, they seem to overlap a lot, yet are declared different. Tukey is a very conservative mean separation test, and the results shown are very liberal. Also, this is properly a two-way ANOVA (trt X location on branch), yet it seems to have been analyzed as a one-way (only trt).
Finally, the authors should cut out repetitive phrases. show/showed/shows appear 32 times, and are always not needed. Just state the facts, then put the supporting citation (table, figure, or source) in brackets. Similarly, also (4x), in addition (11x) and in this study (7x) are not necessary.
I recommend moderate revision, but there is no recommendation for that, so I put down major revision instead, although it is not that drastic.
Author Response
Reviewer 1
- The senior author has published a number of papers on cherry physiology and quality. Although her most recent publication (Matteo et al., 2022) is listed in the references, it is not cited in the body of the text. I did not make time to check on the other references, but urge her to make sure that they are indeed cited.
A1. We considered your comment and included Matteo et al., (2022) in body of the text lines 83 and 98. Besides we checked all references.
- It is usually considered that size, firmness, sugar, and maybe color are the attributes that consumers look for in cherries. In this study, small fruit were firmer, while big fruit were sweeter. It would have been interesting to have a taste panel determine actual preferences, or at least to cite studies showing what is really important to consumers regarding cherry quality.
A2. It was not the aim of this experiment to study the taste of cherries using a panel. We will take it into account in the future.
The paragraph considering the attributes that consumers look for was eliminated to make the introduction more focused.
- The VPD studies, while interesting, do not have SD or SE bars. The lines are drawn with various curves, but seem to be mostly linear, except for the summary graph. It is not clear if or how the various treatments are affected by VPD or stomatal aperture, or if that affects fruit quality attributes mentioned earlier.
A3. We appreciate the comments of the reviewer 1 about the interesting relationship between stomatal conductance and VPD. Figure 3 shows the best fit for each treatment as well as the 95 % confidence interval (CI), which means that you have a 5 % chance of, under a specific VPD, measuring a stomatal conductance value outside the shaded area. It was calculated as the sample mean plus/minus the 95 % of the sample standard deviation divided by the square root of the sample size. CI = sample mean ± confidence level value * sample standard deviation / Square root (sample size)
We consider that this is the best option to highlight how spread out the data is, compared to the mean value estimated. Some other articles published in Plants have chosen this test to highlight the interaction between two variables (please see below):
Yin et al., 2021. Coupling Relationship of Leaf Economic and Hydraulic Traits of Alhagisparsifolia Shap. in a Hyper-Arid Desert Ecosystem. Plants, 10(9), 1867; https://doi.org/10.3390/plants10091867
Stagg et al., 2022. Presence of the Herbaceous Marsh Species Schoenoplectus americanus Enhances Surface Elevation Gain in Transitional Coastal Wetland Communities Exposed to Elevated CO2 and Sediment Deposition Events. Plants, 11(9), 1259; https://doi.org/10.3390/plants11091259
Fontaine et al., 2022. Niche Variation in Endemic Lilium pomponium on a Wide Altitudinal Gradient in the Maritime Alps. Plants, 11(6), 833; https://doi.org/10.3390/plants11060833
Regarding the linear relationship, we agree with the reviewer 1 in that for TR-1, TR-2, and TR-3, the relationship is similar to a linear fit. However, TR-4 and TR-5 showed a clear logarithmic relationship, in which small increases in VPD strongly decreased the stomatal conductance. The rapid stomatal closure in the branches with 50% restrictions in the xylem when the atmospheric demand increased was explained in section "2.4 Water relations" and was related with high Ca concentration in the fruit. Moreover, the relationship between leaf gas exchange and fruit nutrient profile was discussed in lines 399 - 410.
- The manuscript does not properly follow the Rule of the Decimal Place. If there are no numbers to the left of the decimal point, you can present up to two numbers to the right of the decimal One or two numbers to the left of the decimal allows one digit to the right of the decimal. Three or more digits to the left means no digits to the right of the decimal, and in fact no decimal is needed in that case.
A4. We changed the numbers along the whole manuscript following the Rule of the Decimal Place.
- Figure 1-- why is this presented as a box-plot?
A5. In our opinion a box plot is a good way to show the distribution and skewness of data measured for yield per branch´s segment. However, additional information is provided in the following table to reviewer 1. We will add it in additional material to the readers.
Table n: Fruit yield (g) of individual vertical branches with vascular restrictions, in the scion rootstock combination ‘Lapins’/Colt, trained as Kym Green Bush.
|
|
Yield (g) |
||
|
Treatment |
|
Branch |
|
|
|
Upper segment |
Lower segment |
Total |
|
TR1 - CTL |
749 614 749 724 604 |
826 a 514 ab 256 b 555 ab 409 b |
1,575 a |
|
TR2 - G |
1,129 ab |
||
|
TR3 - G+G |
1,004 b |
||
|
TR4 - S |
1,279 ab |
||
|
TR5 - S+G |
1,013 b |
||
|
p-value |
0.643 |
0.001 |
0.027 |
* Different letters within the same column indicate significant differences among treatments according to ANOVA and Tukey test (p≤0.05) (n = 10). TR1 – CTL: a control branch with neither girdling (G) nor transverse incision cutting 50% of xylem cross–section area (S), TR2 – G: a branch with G at its base, TR3 – G+G: a branch with G at its base and G further up at the change of year between the second and the third years of growth, TR4 – S: a branch with S at its base and TR5 – S + G: a branch with S and G at its base. Lower segment from the crown to 1.2 m height on the branch. Upper segment from 1.2 m to 2.4 height on the branch.
- Figure 2 Are these SD or SE bars? If these are Tukey mean separations, they seem to overlap a lot, yet are declared different. Tukey is a very conservative mean separation test, and the results shown are very liberal. Also, this is properly a two-way ANOVA (trt X location on branch), yet it seems to have been analyzed as a one-way (only trt).
A6. Figure 2 shows the mean values and standard deviation of the dry weight of fruit harvested from each segment and treatment. We have included that information in the description of Figure 2 to make clearer the figure and avoid confusion for the reader.
As the reviewer 1 pointed out, when standard error bars for the groups overlap, and the sample sizes are equal, we can state that the difference between the two means is not statistically significant (p > 0.05). However, as our error bars represent standard deviation rather than standard error, then no conclusion is possible.
With the aim of analyzing the effect of the treatments imposed, we did a one-way ANOVA and a Tukey test to assess how vascular restrictions in the branches affect the fruit dry weight produced per branch. However, we did not work with treatment or segment as a factor but both, treatment and segment, so there were 10 groups analyzed: TR-1 upper, TR-1 lower, TR-2 upper, TR-2 lower, TR-3 upper, TR-3 lower, TR-4 upper, TR-4 lower, TR-5 upper and TR-5 lower.
When we compared the yield and the dry weight of the fruit located on each segment within the same treatment (one-way ANOVA, factor location), we observed that there were no significant differences for TR-1, TR-2, TR-4, and TR-5, neither in yield nor in fruit dry weight. However, there were significant differences for TR-3, the upper segment had a significantly higher yield and dry weight than the lower segment.
We agree with the reviewer that a two-way ANOVA (treatment x location on branch) would have been the best option to analyze the data if we had not worked with TR-3, and the location on the branch would not be differently affected by the treatments. However, the differences measured in the yield and fruit dry weight of TR-3 cannot be exclusively related to the location of the segment but also to the double girdling. We chose a one-way ANOVA with treatment-segment factor and 10 groups to compare both segments of TR-3 with the other treatments. Thus, Figure 2 shows no significant differences between segments for all treatments except for TR-3.
- Finally, the authors should cut out repetitive phrases. show/showed/shows appear 32 times, and are always not needed. Just state the facts, then put the supporting citation (table, figure, or source) in brackets. Similarly, also (4x), in addition (11x) and in this study (7x) are not necessary.
A7. We have erased and/or replaced the not necessary words such as show, similarly an so on..
- I recommend moderate revision, but there is no recommendation for that, so I put down major revision instead, although it is not that drastic.

Reviewer 2 Report
Dear Authors
The topic of the article is interesting. However, some editing is required before the article may be published. Reading the abstract, it appears that the article is about carbohydrates, whereas in fact the objective is girdling. It is advised that the abstract be rewritten.
Furthermore, the introduction lacks the necessary continuity and focuses on calcium. The importance of calcium and carbohydrates after cherry harvesting is addressed in the final paragraph of the introduction. it is advised that the hypothesis of this plan be expressed more clearly and explicitly.
Author Response
Reviewer 2
- The topic of the article is interesting. However, some editing is required before the article may be published. Reading the abstract, it appears that the article is about carbohydrates, whereas in fact the objective is girdling. It is advised that the abstract be rewritten.
- Furthermore, the introduction lacks the necessary continuity and focuses on calcium. The importance of calcium and carbohydrates after cherry harvesting is addressed in the final paragraph of the introduction. it is advised that the hypothesis of this plan be expressed more clearly and explicitly.
A1. The suggestions of reviewer 2 have been considered and changes in the abstract and introduction have been made.

Reviewer 3 Report
It is a very complete and interesting study about the effect of phloem and xylem restrictions on fruit quality.
A thorough literature review was conducted.
The experimental design has been clearly defined and explained. However, the M& M section has been moved, and it is not followed by the results section.
Lines 108-109: “The vascular restrictions imposed on ‘Lapins’/Colt sweet cherry vertical branches at eight DAFB had significant effects on the harvest date and fruit yield”. These results must be based on an statistical analysis (Anova analysis or similar), the information about the statistical analysis carried out should be added.
Lines 173-175: “…the fruit from the upper segment showed the significantly highest (p = 0.0001) average fruit fresh weight (FW, 11.7 g fruit-1; Table 2) compared to the other treatments (from 8.1 to 9.8 g fruit-1; Table 2)”. The type of statistical analysis carried out should be defined.
Lines 178-180: “Regarding the total fruit dry weight (DW), the branches with a cross–sectional cut to 178 the xylem and girdling at their base (TR5 – S+G), had lower fruit DW values (p < 0.0001), in both upper and lower segments, compared to the other treatments, which had similar values (Table 2)”. The type of statistical analysis carried out should be defined.
Lines 194-195: “The fruit diameter was significantly (p < 0.0001) affected by the vascular restrictions imposed (Table 2)”. In the same lines as the comments above, The type of statistical analysis carried out should be defined.
Lines 205-206: “However, no significant differences (p 205 = 0.06) in fruit´s firmness (71.5 shore units) were detected in the lower segment of branches 206 among treatments (Table 2). The discussion about this result could be extended.
Due to the objective of the study, the results about fruit firmness, soluble solid content and total acidity could be remarked in the result and conclusion section.
Figure 5: the correlation matrix should be a table.
In figure 5 not all the variables are considered in the correlation matrix. It would be very interesting to include all the variables measured and studied.
Figure 5: the fruit quality variables (weight, firmness, soluble solid content and acidity) should be included.
Considering the high number of interrelated variables studied (some variables are affected by others), another statistical analysis could be added. A principal component analysis (or other analysis) could be addressed to summarize and group the information obtained.
The discussion is extensive, but it could be confusing, organizing and summarizing the information discussed could be adequate.
The conclusions are clear and concise.
Author Response
Reviewer3
- It is a very complete and interesting study about the effect of phloem and xylem restrictions on fruit quality.
- A thorough literature review was conducted.
- The experimental design has been clearly defined and explained. However, the M& M section has been moved, and it is not followed by the results section.
A1. The journal is required to place the M&M section after the discussion.
- Lines 108-109: “The vascular restrictions imposed on ‘Lapins’/Colt sweet cherry vertical branches at eight DAFB had significant effects on the harvest date and fruit yield”. These results must be based on statistical analysis (Anova analysis or similar), the information about the statistical analysis carried out should be added.
A2. For further information we added table N to explain the reviewer the statistical results.
Tabla n: Fruit yield (g) of individual vertical branches with vascular restrictions, in the scion rootstock combination ‘Lapins’/Colt, trained as Kym Green Bush.
|
|
Yield (g) |
||
|
Treatments |
|
Branch |
|
|
|
Upper segment |
Lower segment |
Total |
|
TR1 - CTL |
749 614 749 724 604 |
826 a 514 ab 256 b 555 ab 409 b |
1,575 a |
|
TR2 - G |
1,129 ab |
||
|
TR3 - G+G |
1,004 b |
||
|
TR4 - S |
1,279 ab |
||
|
TR5 - S+G |
1,013 b |
||
|
p-value |
0.643 |
0.001 |
0.027 |
* Different letters within the same column indicate significant differences among treatments according to ANOVA and Tukey test (p≤0.05) (n = 10). TR1 – CTL: a control branch with neither girdling (G) nor transverse incision cutting 50% of xylem cross–section area (S), TR2 – G: a branch with G at its base, TR3 – G+G: a branch with G at its base and G further up at the change of year between the second and the third years of growth, TR4 – S: a branch with S at its base and TR5 – S + G: a branch with S and G at its base. Lower segment from the crown to 1.2 m height on the branch. Upper segment from 1.2 m to 2.4 height on the branch.
- Lines 173-175: “…the fruit from the upper segment showed the significantly highest (p = 0.0001) average fruit fresh weight (FW, 11.7 g fruit-1; Table 2) compared to the other treatments (from 8.1 to 9.8 g fruit-1; Table 2)”. The type of statistical analysis carried out should be defined.
A3. Please, see the statistical description at the bottom of Table 2.
- Lines 178-180: “Regarding the total fruit dry weight (DW), the branches with a cross–sectional cut to 178 the xylem and girdling at their base (TR5 – S+G), had lower fruit DW values (p < 0.0001), in both upper and lower segments, compared to the other treatments, which had similar values (Table 2)”. The type of statistical analysis carried out should be defined.
A4. Please, see the statistical description at the bottom of Table 2.
- Lines 194-195: “The fruit diameter was significantly (p < 0.0001) affected by the vascular restrictions imposed (Table 2)”. In the same lines as the comments above, The type of statistical analysis carried out should be defined.
A5. Please, see the statistical description at the bottom of Table 2.
- Lines 205-206: “However, no significant differences (p 205 = 0.06) in fruit´s firmness (71.5 shore units) were detected in the lower segment of branches 206 among treatments (Table 2). The discussion about this result could be extended.
A3. We added the explanation (lines 206-207) of finding no significant difference in fruit firmness in the lower section.
- Due to the objective of the study, the results about fruit firmness, soluble solid content and total acidity could be remarked in the result and conclusion section.
A4. We added more information about fruit firmness, soluble solid content and total acidity in the discussion and conclusions.
- Figure 5: the correlation matrix should be a table. Figure 5: the fruit quality variables (weight, firmness, soluble solid content and acidity) should be included.
A6. We changed the correlation matrix for a table.
- Considering the high number of interrelated variables studied (some variables are affected by others), another statistical analysis could be added. A principal component analysis (or other analysis) could be addressed to summarize and group the information obtained.
A5. Please see A6.
In Figure 1 attached initially included in the article we only focused on those with significantly different correlations, but we worked on Figure 2 as suggested by the reviewer 3 (see Figure 2 attached).
As the reviewer pointed out, we also worked on a principal component analysis to summarize the information.
However, when we grouped the data, no clear results were obtained, and all treatments overlapped (see Figure 3 attached).
Figure 1. Heat–map of the correlation matrix (Pearson correlation coefficient) between the leaf area to fruit (LA:F) ratio, Ca concentration (Ca), stoichiometric ratios (N:Ca, K:Ca and Mg:Ca) and fruit´s firmness in vertical branches of the sweet cherry fruit combination ‘Lapins’/Colt, trained as Kym Green Bush, at harvest. Values marked *, ** and *** are significant at p < 0.05, p < 0.01 and p < 0.001, respectively.
Figure 2. Heat–map of the correlation matrix (Pearson correlation coefficient) between the leaf area to fruit (LA:F) ratio, Ca concentration (Ca), stoichiometric ratios (N:Ca, K:Ca and Mg:Ca), unitary fresh weight, fruit´s firmness, TSS and TA in vertical branches of the sweet cherry fruit combination ‘Lapins’/Colt, trained as Kym Green Bush, at harvest. Values marked *, ** and *** are significant at p < 0.05, p < 0.01 and p < 0.001, respectively.
|
|
|
Figure 3. Principal component analysis (PCA) biplot of the physicochemical properties of fruit obtained from vertical branches of the sweet cherry fruit combination ‘Lapins’/Colt, trained as Kym Green Bush, at harvest. TR1 – CTL: a control branch with neither girdling (G) nor transverse incision cutting 50% of xylem cross–section area (S), TR2 – G: a branch with G at its base, TR3 – G+G: a branch with G at its base and G further up at the change of year between the second and the third years of growth, TR4 – S: a branch with S at its base and TR5 – S + G: a branch with S and G at its base. Lower segment from the crown to 1.2 m height on the branch. Upper segment from 1.2 m to 2.4 height on the branch.
- The discussion is extensive, but it could be confusing, organizing and summarizing the information discussed could be adequate.
A6. The discusión was complemented and organized to make it clearer, but its is hard to abbreviatte it so much.
- The conclusions are clear and concise.
PLEASE SEE FIGURES IN THE END OF ATTACHED MANUSCRIPT.

Round 2
Reviewer 1 Report
The research described was done thoroughly, but only for one year. The authors cite numerous papers on girdling, most of which find positive results, and almost all of which repeated the experiments for at least two seasons. This is particularly important to determine effects of girdling on return bloom, which the authors did not even mention. Articles they cite mention that there was no, or a positive, effect on return bloom. On the other hand, the authors point out that they did not get increased yield from girdling, so who knows what effect they would find on return bloom. This is a major flaw in the research reported. If they can supply these data, they should add them to a resubmitted article.
The title does not read smoothly and could be rewritten: "Vascular restriction affects composition and quality of sweet cherry fruit"
Other comments, before I got concerned about return bloom:
line 11 carbohydrate (small c)
55 transpiratory--> transpiration, in all cases in the manuscript
97 elucidated
98 poulations (check for Hispanic spelling in other places in the manuscript)
100 not enough
159 drop "the ANOVA" in all table captions. You can't do Tukey test without doing ANOVA, and you do not report any ANOVA numbers in any event (which is fine).
365 I looked up "shore unit", since I am used to fruit firmness being reported in N (newtons). Shore unit is used for firmness of dry materials. The authors do not mention how they measured fruit firmness, but if they used an Effegi device or an Instron or similar, the units should be reported in N.
427 affected
Table 3 and elsewhere. No need to report mg/fruit; mg/100 gdw is fine alone.
339 and elsewhere Just report number of citation, as you do in most of the manuscript, not author's name and date plus citation number. Rephrase sentences accordingly.
342 Despite that,
As opposed to my previous recommendation of "moderate revision", I now request a major revision-- addition of return bloom data. If the data are not available, then the experimental results cannot be regarded as complete and the paper should not be accepted.
Author Response
Dear Reviewer 1,
We apreciatte your comments and suggestions to improve the final version of the manuscript.
We agree with you in the importance of measuring the effects of girdling on different reproductive variables such as the return bloom, however, in this study we did not include it in the hypothesis and objectives. Our aims was to use vascular restrictions as experimental tool to manipulate fruit’s nutritional profile and quality in sweet cherry trees during fruit development. Although, we will take it into account in future experiments in sweet cherry.
We have reviewed several papers that worked with vascular restrictions and not all of them include data from several seasons. Here, there are some examples of recent publications that were focused on one season only, as we did. Please, see below:
Yang, X.-Y.; Wang, F.-F.; Teixeira da Silva, J.A.; Zhong, J.; Liu, Y.-Z.; Peng, S.-A. 2013. Branch girdling at fruit green mature stage affects fruit ascorbic acid contents and expression of genes involved in l-galactose pathway in citrus. New Zealand Journal of Crop and Horticultural Science. 41:1, 23-31.
Pasqualotto, G.; Carraro, V.; De Gregorio, T.; Suarez Huerta, E.; Anfodillo, T. 2019. Girdling of fruit-bearing branches of Corylus avellana reduces seed mass while defoliation does not. Scientia Horticulturae. 255, 37-43.
Pereira, G.E.; Padhi, E.M.T.; Girardello R.C.; Medina-Plaza, C.; Tseng, D.; Bruce, R.C.; Erdmann, J.N.; Kurtural, S.K.; Slupsky C.M.; Oberholster, A. 2020. Trunk Girdling Increased Stomatal Conductance in Cabernet Sauvignon Grapevines, Reduced Glutamine, and Increased Malvidin-3-Glucoside and Quercetin-3-Glucoside Concentrations in Skins and Pulp at Harvest. Front. Plant Sci. 11:707.
Ghadage, N.J.; Patil, S.J.; Khopade, R.Y.; Hiray, S.A.; Patel, B.B. 2021. Effect of time and width of girdling on fruit quality of mango (Mangifera indica L.) cv. Alphonso. International Journal of Chemical Studies. 7(2), 40-42.
Lo Piccolo, E.; Aranti, F.; Landi, M.; Massai, r.; Guidi, L.; Aberavoli, M.R.; Remorini, D. 2021. Girdling stimulates anthocyanin accumulation and promotes sugar, organic acid, amino acid level and antioxidant activity in red plum: An overview of skin and pulp metabolomics. Scientia Horticulturae. 280, 109907.
Winkler, A.; Knoche, M. 2021. Xylem, phloem and transpiration flows in developing European plums. Plos One. 16(5): e0252085.
Ulker, T.; Kamiloglu, M.U. 2021. Influences of girdling and potassium treatments on fruit quality and some physiological characters of ‘Fremont’ mandarin variety. Folia Hort. 33(1), 195-202.
Christopoulos, M.V.; Kafkaletou, M.; Karantzi, A.D.; Tsantili, E. 2021. Girdling Effects on Fruit Maturity, Kernel Quality, and Nutritional Value of Walnuts (Juglans regia L.) alongside the Effects on Leaf Physiological Characteristics. Agronomy. 11, 200.
Rana, V.S.; Zarea, S.E.; Sharma, S.; Rana, N; Kumar, V.; Sharma, U. 2022. Differential Response of the Leaf Fruit Ratio and Girdling on the Leaf Nutrient Concentrations, Yield, and Quality of Nectarine. Journal of Plant Growth Regulation.
Wang S.W.; Zhang C.F.; Pan C.D.; Zhao S.C. 2022. Analysis of walnut fruit quality based on source-sink relationships. Hort. Sci. (Prague), 49: 102–108.
With respect to the additional changes you proposed, most of them were accepted and highlighted in yellow; while those that we kept unchanged are with a side comment.
Thank you for your kind help in this process.
Sincerely,
The authors
